# Comparative Pharmacokinetics and Pharmacodynamics of a Novel Sodium-Glucose Cotransporter 2 Inhibitor, DWP16001, with Dapagliflozin and Ipragliflozin

**DOI:** 10.3390/pharmaceutics12030268

**Published:** 2020-03-15

**Authors:** Min-Koo Choi, So Jeong Nam, Hye-Young Ji, Mi Jie Park, Ji-Soo Choi, Im-Sook Song

**Affiliations:** 1College of Pharmacy, Dankook University, Cheon-an 31116, Korea; minkoochoi@dankook.ac.kr; 2College of Pharmacy and Research Institute of Pharmaceutical Sciences, Kyungpook National University, Daegu 41566, Korea; goddns159@nate.com; 3Life Science Institute, Daewoong Pharmaceutical, Yongin, Gyeonggido 17028, Korea; hychi138@daewoong.co.kr (H.-Y.J.); mjpark201@daewoong.co.kr (M.J.P.); jschoi172@daewoong.co.kr (J.-S.C.)

**Keywords:** sodium-glucose cotransporter 2 (SGLT2) inhibitors, DWP16001, kidney distribution, inhibition mode

## Abstract

Since sodium-glucose cotransporter 2 (SGLT2) inhibitors reduced blood glucose level by inhibiting renal tubular glucose reabsorption mediated by SGLT2, we aimed to investigate the pharmacokinetics and kidney distribution of DWP16001, a novel SGLT2 inhibitor, and to compare these properties with those of dapagliflozin and ipragliflozin, representative SGLT2 inhibitors. The plasma exposure of DWP16001 was comparable with that of ipragliflozin but higher than that of dapagliflozin. DWP16001 showed the highest kidney distribution among three SGLT2 inhibitors when expressed as an area under curve (AUC) ratio of kidney to plasma (85.0 ± 16.1 for DWP16001, 64.6 ± 31.8 for dapagliflozin and 38.4 ± 5.3 for ipragliflozin). The organic anion transporter-mediated kidney uptake of DWP16001 could be partly attributed to the highest kidney uptake. Additionally, DWP16001 had the lowest half-maximal inhibitory concentration (IC_50_) to SGLT2, a target transporter (0.8 ± 0.3 nM for DWP16001, 1.6 ± 0.3 nM for dapagliflozin, and 8.9 ± 1.7 nM for ipragliflozin). The inhibition mode of DWP16001 on SGLT2 was reversible and competitive, but the recovery of the SGLT2 inhibition after the removal of SGLT2 inhibitors in CHO cells overexpressing SGLT2 was retained with DWP16001, which is not the case with dapagliflozin and ipragliflozin. In conclusion, selective and competitive SGLT2 inhibition of DWP16001 could potentiate the efficacy of DWP16001 in coordination with the higher kidney distribution and retained SGLT2 inhibition of DWP16001 relative to dapagliflozin and ipragliflozin.

## 1. Introduction

Achieving appropriate glycemic control for type 2 diabetes patients is a prerequisite for preventing cardiovascular and microvascular complications, and this can be guided by a combination of antidiabetic drugs with different modes of action [1]. 

Sodium-glucose cotransporter 2 (SGLT2) inhibitors are the latest class of antidiabetic drugs that act through the inhibition of renal tubular glucose reabsorption and a reduction of blood glucose levels without stimulating insulin release [2]. SGLT2 is expressed in the S1 segment of proximal kidney tubules and is responsible for roughly 90% of the reabsorption of filtered glucose [3]. Additionally, SGLT2 inhibitors exhibit low hypoglycemia risk and have been associated with a significant reduction in major adverse cardiovascular events in clinical trials [4], garnering SGLT2 inhibitors increased attention.

Several SGLT2 inhibitors have been approved for the treatment of type 2 diabetes, including canagliflozin (Invokana^®^), dapagliflozin (Farxiga^®^), empagliflozin (Jardiance^®^), ipragliflozin (Suglat^®^), and tofogliflozin (Apleway^®^) [3,5]. 

Additionally, there are several other similar compounds in the pipeline that may be approved in the near future. DWP16001 (Figure 1), a selective SGLT2 inhibitor, is under development by Daewoong Pharmaceutical Co. Ltd. (Yongin, Korea) and is currently undergoing phase 2 clinical trials (Registration No. NCT04014023). 

Recently, Tahara et al. [5] compared the pharmacokinetics, pharmacodynamics, and pharmacological characteristics of six SGLT2 inhibitors, such as ipragliflozin, dapagliflozin, tofogliflozin, canagliflozin, empagliflozin, and luseogliflozin. The study showed that all the SGLT2 inhibitors induced urinary glucose excretion in a dose-dependent manner but the duration of action differed among the six drugs. Ipragliflozin and dapagliflozin showed persistent duration of action; these two drugs exhibited increased urinary glucose excretion even 18 h post dose but the others showed about 12 h of duration. In addition, ipragliflozin and dapagliflozind showed the lowest blood glucose and insulin level following the same daily dose (3 mg/kg) in diabetic mice. The long duration of action and glucose lowering efficacy of these two drugs closely correlated with the drug distribution and retention in the kidney and elimination half-life [5], suggesting the importance of kidney distribution and elimination profile is prerequisite in the efficacy of SGLT2 inhibitors, as well as the potent SGLT2 inhibition. Therefore, this study aimed to compare the pharmacokinetic properties and kidney distribution of DWP16001 with those of dapagliflozin and ipragliflozin, representative SGLT2 inhibitors that showed potent and long duration efficacy [5] and to compare the selectivity and mode of inhibition of DWP16001 on SGLT2 with dapagliflozin and ipragliflozin to evaluate the potency of DWP16001 over other SGLT2 inhibitors.

## 2. Materials and Methods 

### 2.1. Materials

DWP16001 and D4-DWP16001 (for internal standard (IS)), were obtained from Daewoong Pharmaceutical Co. Ltd. (Yongin, Korea). Dapagliflozin and ipragliflozin were obtained from Toronto Research Chemicals Inc. (North York, ON, Canada) (Figure 1).

Para-aminohipuric acid (PAH), methyl a-D-glucopyranoside (AMG), sodium dodecyl sulfate (SDS), G418, non-essential amino acids, 4-(2-hydroxyethyl)-1-piperazineethanesulfonic acid (HEPES), and Hank’s balances salts solution (HBSS, pH 7.4) were purchased from Sigma–Aldrich Chemical Co. (St. Louis, MO, USA). [^14^C]Methyl-α-D-glucopyranoside (AMG) (290 mCi/mmol) was purchased from Moravek (Brea, CA, USA). [^3^H]Estrone-3-sulfate (ES) (2.12 TBq/mmol) and [^3^H]para-aminohipuric acid (PAH) (0.13 TBq/mmol) were purchased from Perkin Elmer Inc. (Boston, MA, USA). Dulbecco’s modified Eagle’s medium (DMEM), RPMI1640 medium, fetal bovine serum (FBS), and poly-D-lysine-coated 24-well plates were purchased from Corning (Tewksbury, MA, USA). All other chemicals and solvents were reagent or analytical grade. 

CHO cells overexpressing SGLT1 and SGLT2 (CHO-SGLT1 and -SGLT2, respectively) and CHO-mock cells were obtained from Daewoong Pharmaceutical Co. Ltd. (Yongin, Korea). HEK293 cells overexpressing organic anion transporter 1 (OAT1) and OAT3 (HEK293-OAT1 and -OAT3, respectively) and HEK293-mock cells were purchased from Corning (Tewksbury, MA, USA).

### 2.2. Animals and Ethical Approval

Male Institute of Cancer Research (ICR) mice (7–8-weeks-year-old, 30−35 g) were purchased from Samtako Co. (Osan, Korea). Animals were acclimatized for 1 week in an animal facility at Kyungpook National University. Food and water were available ad libitum. All animal procedures were approved by the Animal Care and Use Committee of Kyungpook National University (Approval No. 2016-0138) and carried out in accordance with the National Institutes of Health guidance for the care and use of laboratory animals.

### 2.3. Pharmacokinetic Study

ICR mice were randomly divided into three groups and were fasted for at least 12 h before the oral administration of DWP16001, dapagliflozin, and ipragliflozin but had free access to water. On the day of pharmacokinetic study, the mice received DWP16001, dapagliflozin, or ipragliflozin solution at a dose of 1 mg/kg (dissolved in a mixture of 10% DMSO and 90% saline) using oral gavage. Blood samples were collected at 0.5, 1, 2, 4, 8, 24, 48, and 72 h following the oral administration (1 mg/kg each) of DWP16001, dapagliflozin, or ipragliflozin and centrifuged at 12,000× *g* for 1 min to separate the plasma. An aliquot (30 µL) of each plasma sample was stored at −80 °C until the analysis. Kidney samples were also isolated at 8, 24, 48, and 72 h following the oral administration of DWP16001, dapagliflozin, or ipragliflozin, minced thoroughly, and homogenized with four volumes of saline using tissue grinder. An aliquot (50 µL) of each kidney homogenate sample was stored at −80 °C until the analysis.

For the analysis of SGLT2 inhibitors, aliquots of plasma (30 µL) and kidney homogenate (50 µL) were added to 100 μL of aqueous solution of D4-DWP16001 (IS, 20 ng/mL), and vigorously mixed with 500 μL methyl tert-butyl ether (MTBE) for 15 min. After centrifugation at 16,000 *g* for 5 min, samples were kept for 1 h at −80 °C to make an aqueous layer freeze. An organic upper layer was transferred to a clean tube and evaporated to dryness under a gentle stream of nitrogen. Then, the dried extract was reconstituted in 150 µL of mobile phase, and a 3 µL aliquot of the reconstituent was injected into a liquid chromatography–tandem mass spectrometry (LC–MS/MS) system. 

Pharmacokinetic parameters, such as the area under plasma concentration-time curve from zero to infinity (AUC), were calculated from plasma concentration vs time curves using non-compartment analysis with WinNonlin (version 5.1; Pharsights, Cary, NC, USA). The AUC ratios (i.e., ratios of kidney AUC to plasma AUC) were calculated by dividing the AUC of the three SGLT2 inhibitors in the kidney by the plasma AUC values of the three SGLT2 inhibitors.

### 2.4. Protein Binding

The protein binding of DWP16001, dapagliflozin, and ipragliflozin (1000 ng/mL) in mouse plasma and 20% kidney homogenate was determined using a rapid equilibrium dialysis kit (ThermoFisher Scientific Korea, Seoul, Korea) according to the manufacturer’s instructions. Briefly, 100 μL of mouse plasma and 20% kidney homogenate samples containing 1000 ng/mL of DWP16001, dapagliflozin, or ipragliflozin were added to the sample chamber of a semipermeable membrane (molecular weight cut-off 8000 Da) and 300 μL of phosphate buffered saline (PBS) was added to the outer buffer chamber. Four hours after incubation at 37 °C on a shaking incubator at 300 rpm, aliquots (50 μL) were collected from both the sample and buffer chambers and treated with equal volumes of fresh PBS and blank plasma or blank kidney homogenate, respectively, to match the sample matrices. The matrix-matched sample (100 μL) was added 100 μL of aqueous solution of D4-DWP16001 (IS, 20 ng/mL), and vigorously mixed with 1000 μL MTBE for 15 min. After centrifugation at 16,000 *g* for 5 min, samples were kept for 1 h at −80 °C. An organic upper layer was transferred to a clean tube and evaporated to dryness under a gentle stream of nitrogen. Then, the dried extract was reconstituted in 300 µL of mobile phase and a 3 µL aliquot of the reconstituent was injected into the LC-MS/MS system. 

Plasma protein binding was calculated using the following Equation (1) [6,7].
(1)$$\mathrm{Undiluted}\;\mathrm{free}\;\mathrm{drug}\;\mathrm{fraction}\;\left(\mathrm{fu}\right)=\frac{\mathrm{Drug}\;\mathrm{concentration}\;\mathrm{in}\;\mathrm{buffer}\;\mathrm{chamber}}{\mathrm{Drug}\;\mathrm{concentration}\;\mathrm{in}\;\mathrm{plasma}\;\mathrm{sample}\;\mathrm{chamber}}$$

Kidney protein binding was calculated using the following equations, Equations (2) and (3), and a dilution factor (D as a value of 5) for tissue homogenates was used since we used 20% kidney homogenates [6,7].
(2)$$\mathrm{Diluted}\;\mathrm{free}\;\mathrm{drug}\;\mathrm{fraction}\;\left(\mathrm{fu}\prime \right)=\frac{\mathrm{Drug}\;\mathrm{concentration}\;\mathrm{in}\;\mathrm{buffer}\;\mathrm{chamber}}{\mathrm{Drug}\;\mathrm{concentration}\;\mathrm{in}\;\mathrm{kidney}\;\mathrm{homogenate}\;\mathrm{chamber}},$$
(3)$$\mathrm{Undiluted}\;\mathrm{free}\;\mathrm{drug}\;\mathrm{fraction}\;\left(\mathrm{fu}\right)=\frac{1/D}{\left(\frac{1}{{\mathrm{fu}}^{\prime }}-1\right)+1/D}=\frac{\mathrm{fu}\prime \times 0.2}{1-\mathrm{fu}\prime \times 0.8}.$$

### 2.5. Substrate Specificity of DWP16001, Dapagliflozin, and Ipragliflozin for OAT1 and OAT3

HEK293-mock cells and HEK293 cells overexpressing OAT1 and OAT3 transporters (HEK293-OAT1 and -OAT3, respectively) were seeded in poly-D-lysine-coated 24-well plates at a density of 4 × 10^5^ cells/well and cultured for 24 h in DMEM supplemented with 10% FBS and 5 mM non-essential amino acids at 37 °C in 8% CO_2_ condition.

For each experiment, the growth medium was discarded after 24 h, and the attached cells were washed with pre-warmed HBSS and incubated with pre-warmed HBSS for 20 min at 37 °C. To confirm the functionality of OAT1 and OAT3, we measured the uptake of 0.1 µM [^3^H]PAH and 0.1 µM [^3^H]ES, representative substrates for OAT1 and OAT3, respectively, into in the HEK293-mock cells and HEK293-OAT1 and -OAT3 cells, respectively, for 5 min in the presence and absence of 20 µM probenecid, a typical inhibitor for both OAT1 and OAT3 [8,9]. The cells were then washed three times with 500 μL of ice-cold HBSS immediately after placing the plates on ice. Subsequently, cells were lysed with 10% sodium dodecyl sulfate and mixed with Optiphase cocktail solution overnight. The radioactivity of the cell lysate was measured using a liquid scintillation counter (Microbeta 2; Perkin Elmer Inc., Boston, MA, USA).

The uptake of DWP16001, dapagliflozin, and ipragliflozin (2 µM each) was measured for 5 min at 37 °C in the HEK293-mock cells and HEK293-OAT1 and -OAT3 cells, respectively, in the absence and presence of 20 µM probenecid. For the concentration dependency in the uptake of DWP16001, the uptake of DWP16001 in a concentration range of 0.5–50 μM dissolved in HBSS was measured for 5 min at 37 °C in the mock cells and HEK293-OAT1 and -OAT3 cells, respectively. After 5 min, the cells were washed three times with 500 μL of ice-cold HBSS immediately after placing the plates on ice. Subsequently, the cells were scraped using a cell scraper with 100 μL of PBS, and cell suspensions were transferred to a clean tube, combined with 100 μL of aqueous solution of D4-DWP16001 (IS, 20 ng/mL), and vigorously mixed with 1000 μL MTBE for 15 min. After centrifugation at 16,000 *g* for 5 min, samples were kept for 1 h at −80 °C. An organic upper layer was transferred to a clean tube and evaporated to dryness under a gentle stream of nitrogen. Then, the dried extract was reconstituted in 300 µL of mobile phase and a 3 µL aliquot of the reconstituent was injected into the LC-MS/MS system.

In the concentration-dependent uptake studies, the transporter-mediated uptake of DWP16001 was calculated by the subtraction of the transport rates of DWP16001 into the mock cells from those of the HEK293-OAT1 and -OAT3 cells. Kinetic parameters for the OAT1- and OAT3-mediated transport of DWP16001 were determined using the Michaelis-Menten equation [*V* = *V*_max_⋅S/(*K*_m_ + S)] [10].

### 2.6. LC-MS/MS Analysis of DWP16001, Dapagliflozin, and Ipragliflozin

Concentrations of DWP16001, dapagliflozin, and ipragliflozin in plasma and kidney homogenate samples were analyzed using an Agilent 6470 triple quadrupole LC–MS/MS system (Agilent, Wilmington, DE, USA).

DWP16001, dapagliflozin, and ipragliflozin were separated on a Synergi Polar RP column (2.0 × 150 mm, 4 μm particle size; Phenomenex, Torrence, CA) using a mobile phase consisting of water (15%) and methanol (85%) containing 0.1% formic acid at a flow rate of 0.25 mL/min.

Quantification of a separated analyte peak was performed at *m*/*z* 464 → 131 for DWP16001 (T_R_ (retention time) 2.8 min), *m*/*z* 422 → 151 for ipragliflozin (T_R_ 2.5 min), *m*/*z* 426 → 167 for dapagliflozin (T_R_ 2.5 min), *m*/*z* 468 → 135 for D4-DWP16001 (T_R_ 2.8 min), in the positive ionization mode with a collision energy (CE) of 25 eV. The calibration standards of a mixture of DWP16001, dapagliflozin, and ipragliflozin in mouse plasma were 5–1000 ng/mL, and intraday and interday precision and accuracy were less than 14.7% in all samples. The calibration standards of a mixture of DWP16001, dapagliflozin, and ipragliflozin in mouse kidney homogenate were 5–1000 ng/mL, and intraday and interday precision and accuracy were less than 13.8% in all samples.

### 2.7. Inhibitory Effects of DWP16001, Dapagliflozin, and Ipragliflozin on the SGLT1 and SGLT2 Activities

CHO cells overexpressing SGLT1 and SGLT2 cells (CHO-SGLT1 and -SGLT2) and CHO-mock cells were characterized as previously described [11]. Cells were maintained in RPMI1640 medium supplemented with 10% fetal bovine serum and 200 µg/mL G418 at 37 °C in 5% CO_2_ conditions. CHO-SGLT1 and -SGLT2 cells were seeded at a density of 1 × 10^5^ cells/well in 96-well plates. After 24 h, the growth medium was discarded from the cells, and the cells were washed with pre-warmed Na^+^-free buffer (10 mM HEPES, 5 mM Tris, 140 mM choline chloride, 2 mM KCl, 1 mM CaCl_2_, 1 mM MgCl_2_, pH7.4) and incubated for 1 h in Na^+^-free buffer. After replacing Na^+^-free buffer with Na^+^ gradient buffer (10 mM HEPES, 5 mM Tris, 140 mM NaCl, 2 mM KCl, 1 mM CaCl_2_, 1 mM MgCl_2_, pH7.4) containing 10 µM [^14^C]AMG, the uptake of [^14^C]AMG into the CHO-mock cells and CHO-SGLT1 and -SLGT2 cells was measured for 0.5, 1, 1.5, 2, and 3 h. After a predetermined incubation time, cells were washed three times with 200 µL of ice-cold Na^+^-free buffer immediately after placing the plates on ice. Then, the cells were lysed with 10% SDS, and the cell lysates were mixed with Optiphase cocktail solution. Thereafter, the radioactivity of the cell lysates was measured using a liquid scintillation counter.

The inhibitory effect of known inhibitors, such as dapagliflozin and ipragliflozin, on [^14^C]AMG uptake in the CHO-mock cells and CHO-SGLT1 and -SLGT2 cells was measured in the presence or absence of dapagliflozin and ipragliflozin (1, 10 μM for SGLT1; 10, 100 nM for SGLT2) for 2 h. For the calculation of IC_50_ values, the uptake of 10 µM [^14^C]AMG in the CHO-mock cells and CHO-SGLT1 and -SLGT2 cells was measured for 2 h with or without DWP16001, dapagliflozin, or ipragliflozin (1 nM–50 µM for SGLT1; 0.01 nM–1 µM for SGLT2). After 2 h, cells were washed three times with 200 µL of ice-cold Na^+^-free buffer and the cells were lysed with 10% SDS (40 µL), followed by adding Optiphase cocktail solution (200 µL). The radioactivity of the cell lysates was measured using a liquid scintillation counter. The SGLT1 or SGLT2-mediated uptake of [^14^C]AMG was calculated by the subtraction of the uptake rates of [^14^C]AMG into the mock cells from those of the CHO-SGLT1 and -SLGT2 cells. In the inhibition studies, the percentages of the transport rate of AMG with or without SGLT2 inhibitors were calculated and the data were fitted to an inhibitory effect model. The IC_50_ (the concentration of the inhibitor to show half-maximal inhibition) values were calculated using Sigma Plot ver.10.0 (Systat Software, Inc.; San Jose, CA, USA) [12].

To investigate time dependency in the inhibition of SGLT2, CHO-mock and -SGLT2 cells were seeded at a density of 1 × 10^5^ cells/well in 96-well plates. After 24 h, the growth medium was discarded from the cells, and the cells were washed with pre-warmed Na^+^-free buffer and pre-incubated with Na^+^-free buffer containing various concentrations of DWP16001, dapagliflozin, or ipragliflozin (0.001 nM–100 nM) for 1 and 2 h. Then, after replacing Na^+^-free buffer with Na^+^ gradient buffer containing 10 µM [^14^C]AMG and various concentrations of DWP16001, dapagliflozin, or ipragliflozin (0.001 nM–100 nM), the uptake of [^14^C]AMG into the CHO-mock and -SGLT2 cells was measured for 2 h. After 2 h of incubation, the radioactivity of the cell lysate was measured following the same sample preparation method described above.

To investigate the mode of inhibition of the three SGLT2 inhibitors, the inhibition experiments were initiated by replacing Na^+^-free buffer with Na^+^ gradient buffer containing 1, 2.5, 5, and 50 µM [^14^C]AMG and various concentrations of DWP16001, dapagliflozin, or ipragliflozin (0.001 nM–250 nM) and the uptake of [^14^C]AMG into the CHO-mock and -SGLT2 cells was measured for 2 h. The radioactivity of the cell lysate was measured following the same sample preparation method described above. Uptake rate of AMG and concentrations of DWP16001, dapagliflozin, or ipragliflozin were plotted to Dixon plots to identify the mode of inhibition [13,14].

To investigate the recovery of SGLT2 activity depending on the washout period after 24 h exposure of DWP16001, dapagliflozin, or ipragliflozin, CHO-SGLT2 cells were seeded at a density of 1 × 10^5^ cells/well in 96-well plates. After 24 h, the growth medium was discarded from the cells, and the cells were treated with RPMI1640 medium containing DWP16001, dapagliflozin, or ipragliflozin (0.2, 2, 20, and 200 nM) for 24 h. After 24 h, the RPMI1640 medium was replaced with pre-warmed fresh RPMI1640 medium and incubated for 1, 2, 3, 5, and 6 h, and proceeded another pre-incubation with Na^+^-free buffer for 1 h. Then, after replacing Na^+^-free buffer with Na^+^ gradient buffer containing 10 µM [^14^C]AMG, the uptake of [^14^C]AMG into the CHO-SGLT2 cells was measured for 2 h. After 2 h of incubation, the radioactivity of the cell lysate was measured following the same sample preparation method described above.

### 2.8. Statistics

The statistical significance was assessed by t-test using SPSS for Windows (version 24.0; IBM Corp., Armonk, NY, USA).

## 3. Results

### 3.1. LC-MS/MS Analysis of DWP16001, Dapagliflozin, and Ipragliflozin

To compare the pharmacokinetics and tissue distribution of DWP16001, dapagliflozin, ipragliflozin in mice, analyses of the three SGLT2 inhibitors using LC-MS/MS were applied. Figure 2 shows the selected precursor and product ions of DWP16001, dapagliflozin, ipragliflozin, and D4-DWP16001 (IS). The selected precursor and product ions of dapagliflozin and ipragliflozin were consistent with previously published findings [12,15].

Representative multiple reaction-monitoring (MRM) chromatograms of DWP16001, D4-DWP16001 (IS), dapagliflozin, and ipragliflozin (Figure 3) showed that all the analyte peaks obtained using the liquid-liquid extraction method using MTBE were well separated with no interfering peaks at their respective retention times.

### 3.2. Pharmacokinetics and Kidney Distribution of DWP16001, Dapagliflozin, and Ipragliflozin in Mice

To compare the pharmacokinetic profile of DWP16001 with those of dapagliflozin and ipragliflozin, the same dose of the three SGLT2 inhibitors was orally administered to ICR mice (1 mg/kg each), and the concentrations of DWP16001, dapagliflozin, and ipragliflozin in plasma and kidney samples were analyzed. The PK profiles of DWP16001, dapagliflozin, and ipragliflozin are shown in Figure 4 and the PK parameters were summarized in Table 1.

As shown in Figure 4, plasma concentrations of DWP16001 were similar to those of ipragliflozin and greater than those of dapagliflozin. Consequently, the AUC and C_max_ values of DWP16001 were similar to those of ipragliflozin and higher than those of dapagliflozin. However, the concentrations of DWP16001 in the kidney were maintained at higher concentrations for 72 h compared with those of ipragliflozin and dapagliflozin, suggesting the prolonged efficacy of DWP16001 via the inhibition of SGLT2, which is located in the renal proximal tubule [3]. Because of the high and prolonged concentration of DWP16001, the AUC, AUC ratio, and t_1/2_ values of DWP16001 were significantly greater than those of dapagliflozin and ipragliflozin (Table 1).

### 3.3. Substrate Specificity of DWP16001, Dapagliflozin, and Ipragliflozin for OAT1 and OAT3

The AUC ratios of these three SGLT2 inhibitors were much greater than unity, suggesting that DWP16001, dapagliflozin, and ipragliflozin are distributed in the kidney, in which tissue protein binding ability to the specific protein or drug transporters may be involved. Therefore, firstly, we measured the protein binding of DWP16001, dapagliflozin, and ipragliflozin in the plasma and kidney homogenate.

The plasma and kidney tissue protein binding of DWP16001, dapagliflozin, and ipragliflozin were high and comparable and the logP values of three compound were also comparable (i.e., in the range of 2.27–2.65) (Table 2). It suggests that the large distribution of DWP16001, dapagliflozin, and ipragliflozin in the kidney was not necessarily reliant on plasma/tissue-specific binding ability or lipophilicity of these compounds.

We conducted further investigations into the tissue-specific transport of DWP16001, dapagliflozin, and ipragliflozin by using cell systems overexpressing OAT1 and OAT3 since OAT1 and OAT3 are exclusively distributed in the kidney and contribute to the drug distribution to the kidney [16]. The functionality of the OAT1 and OAT3 transport system was confirmed by the significantly greater uptake rates of the probe substrates in HEK293-OAT1 and -OAT3 cells than that in HEK293-mock cells (i.e., 14.7-fold increase in PAH uptake for OAT1 and 16.4-fold increase in ES uptake for OAT3) and by the decreased uptake rate of each probe substrate by the addition of a representative inhibitor, probenecid (Figure 5A,B) [17]. The uptake of DWP16001 in HEK293-OAT1 and OAT3 cells was significantly higher than that in HEK293-mock cells, and it was significantly decreased by the presence of probenecid (Figure 5C). However, dapagliflozin and ipragliflozin were not substrates for both OAT1 and OAT3 (Figure 5D,E). Next, we investigated the concentration dependency in the OAT1 and OAT3-mediated DWP16001 uptake. As shown in the figure, the OAT1 and OAT3-mediated uptake of DWP16001 showed saturable kinetics, and the kinetic parameters, such as K_m_ and V_max_, calculated from the concentration-dependent uptake of DWP16001 in HEK293-OAT1 and –OAT3 cells are shown in Figure 6. Collectively, OAT1 and OAT3-mediated uptake of DWP16001 could contribute to the highest kidney distribution of DWP16001 among three SGLT2 inhibitors. However, these processes were not dominant over the cellular uptake of the three SGLT2 inhibitors into HEK293-mock cells (i.e., passive diffusion), which is evident from two-fold increases in HEK293-OAT1 or -OAT3 cells compared with mock cells (Figure 5C). Therefore, OAT1- and OAT3-mediated active transport and passive diffusion all contributed to the highly permeable absorptive process of DWP16001 in the kidney, and dapagliflozin and ipragliflozin also showed higher uptake levels into HEK293 cells via passive diffusion.

### 3.4. Inhibition Potential of DWP16001, Dapagliflozin, Ipragliflozin on SGLT1 and SGLT2 Activities

#### 3.4.1. Comparison of IC_50_ Values of DWP16001, Dapagliflozin, and Ipragliflozin on SGLT1 and SGLT2

To confirm the functionality of CHO-SGLT1 cells, we measured the time-dependent uptake of AMG, a representative substrate for SGLT1 [11,18]. As shown in Figure 7A, AMG uptake was increased with incubation time, and the optimal incubation time was selected as 2 h from the linear phase of the slope. AMG uptake into CHO-SGLT1 cells was 42-fold greater than that into CHO-mock cells and significantly inhibited by the presence of known inhibitors, such as dapagliflozin and ipragliflozin [3,5] (Figure 7B). With the same experimental condition, the inhibitory effect of DWP16001, dapagliflozin, and ipragliflozin on the AMG uptake into CHO-SGLT1 cells was measured in a concentration range of 1–50,000 nM and IC_50_ values were calculated (Figure 7C–E; Table 3).

Similarly, the functionality of CHO-SGLT2 cells was confirmed from the time-dependent uptake of AMG, a representative substrate for SGLT2 [3,5]. As shown in Figure 8A, AMG uptake was increased with incubation time, and the optimal incubation time was selected as 2 h from the linear phase of the slope. AMG uptake into CHO-SGLT1 cells was 9.8-fold greater than that into CHO-mock cells and significantly inhibited by the presence of known inhibitors, such as dapagliflozin and ipragliflozin [3] (Figure 8B). With the same experimental conditions, the inhibitory effect of DWP16001, dapagliflozin, and ipragliflozin on the AMG uptake into CHO-SGLT2 cells was measured in a concentration range of 0.001–100 nM, and IC_50_ values were calculated (Figure 8C–E; Table 3).

DWP16001 showed greater affinity for SGLT2 compared with dapagliflozin and ipragliflozin. Moreover, the affinity to SGLT1 was also greater with DWP16001 compared with dapagliflozin and ipragliflozin. When compared with selectivity, which was calculated from the SGLT1/SGLT2 IC_50_ ratio, the selectivity of DWP16001 was high relative to that of dapagliflozin and ipragliflozin (Table 3).

#### 3.4.2. Mode of Inhibition

To investigate whether the inhibition of SGLT2 was time-dependent, the inhibitory effects of DWP16001, dapagliflozin, and ipragliflozin on SGLT2-mediated uptake of AMG were measured with or without pretreatment with DWP16001, dapagliflozin, and ipragliflozin, respectively. Figure 9 shows that the inhibitory potential of DWP16001, dapagliflozin, and ipragliflozin on SGLT2 did not change with different pretreatment times, suggesting that all the DWP16001, dapagliflozin, and ipragliflozin inhibited SGLT2 function in a time-independent manner.

To determine the mode of inhibition on SGLT2, the inhibitory effects of DWP16001, dapagliflozin, and ipragliflozin on the AMG uptake into CHO-SGLT2 cells were measured with different substrate concentrations, and the results were expressed as Dixon plots (Figure 10A–C) and a replot of the Dixon slopes (Figure 10D–F) [13,19,20]. Dixon transformation of inhibition profile by DWP16001 (0.001-250 nM) for different concentrations of AMG indicated competitive inhibition of DWP16001 on SGLT2 activity (Figure 10A). A replot of the Dixon slopes vs 1/S produced a straight line converging on the zero (Figure 10D), suggesting competitive and reversible inhibition of DWP16001. Similar results were shown in case of dapagliflozin and ipragliflozin (Figure 10B,C,E,F). The results collectively suggested that the inhibition of SGLT2 by DWP16001, dapagliflozin, and ipragliflozin is reversible and competitive.

#### 3.4.3. Retained Inhibition Potential of DWP16001 Compared to Dapagliflozin and Ipragliflozin

The recovery of SGLT2 activities depending on the washout period after 24 h exposure of DWP16001, ipragliflozin, and dapagliflozin in CHO-SGLT2 cells was investigated. As shown in Figure 11, the activity of reduced SGLT2 by treatment with DWP16001, dapagliflozin, and ipragliflozin for 24 h in CHO-SGLT2 cells revealed differences in the recovery of SGLT2 depending on the time of drug removal from the medium, as well as SGLT2 inhibitors. In the case of DWP16001, the activity tended to gradually recover as the washout time increased. When treated at a high concentration (200 nM), the activity did not recover even 8 h after the drug was removed from the medium (Figure 11A). However, when dapagliflozin and ipragliflozin were used, SGLT2 activity was recovered to the control level 4 h after the removal of dapagliflozin and ipragliflozin (Figure 11B,C). The results indicate that the dissociation of DWP16001 to SGLT2 is slower and more incomplete than that of dapagliflozin and ipragliflozin and that the binding affinity to SGLT2 is stronger than that of dapagliflozin and ipragliflozin.

## 4. Discussion

DWP16001 is a candidate SGLT2 inhibitor that is currently under development. As a first step, the pharmacokinetic properties and in vitro SGLT2 inhibition were compared with currently used SGLT2 inhibitors. Dapagliflozin and ipragliflozin were selected based on their high kidney distributions and long elimination half-lives (t_1/2_) in the kidney [5], which are thought to be important for the efficacy and duration of action of SGLT2 inhibitors. DWP16001 showed higher kidney distributions compared with dapagliflozin and ipragliflozin. DWP16001 also had longer t_1/2_ in the kidney than dapagliflozin and ipragliflozin, as well as a comparable plasma profile with ipragliflozin (Table 1). Moreover, the kidney concentration of DWP16001 was maintained over 72 h following oral administration of 1 mg/kg DWP16001 (Figure 4). Taken together, these pharmacokinetic results show that the duration of action of DWP16001 is greater than that of dapagliflozin and ipragliflozin and that the oral therapeutic dose of DWP16001 could be reduced compared with both dapagliflozin and ipragliflozin.

To investigate the underlying mechanisms of highest kidney distribution and maintained concentration of DWP16001 in the kidney, we measured the kidney tissue binding and involvement of OAT1 and OAT3 transporters of the three SGLT2 inhibitors. All three SGLT2 inhibitors showed high protein binding, but the kidney tissue binding performances of these three SGLT2 inhibitors were not different from their plasma protein binding capabilities (Table 2). However, DWP16001 was a substrate for both OAT1 and OAT3, which are dominantly expressed in the kidney, whereas dapagliflozin and ipragliflozin were not (Figure 5 and Figure 6). Although this could not solely explain the high kidney distribution, OAT1- and OAT3-mediated transport process may contribute to the high kidney distribution.

Next, we compared the in vitro SGLT2 inhibition and selectivity of SGLT2 inhibition over SGLT1. All three SGLT2 inhibitors inhibited SGLT2 and SGLT1 activity in a concentration-dependent manner and IC_50_ values of dapagliflozin and ipragliflozin were in the range of previous reports (1.0–1.3 nM for dapagliflozin; 6.75–8.07 nM for ipragliflozin) [21,22]. IC_50_ values of DWP16001 to SGLT2 and SGLT1 were lower than those of dapagliflozin and ipragliflozin, suggesting a greater affinity to SGLT2 inhibition for DWP16001 with a higher selectivity over SGLT1 than dapagliflozin and ipragliflozin. The mode of inhibition of DWP16001 was not different from the other SGLT2 inhibitors. They all showed reversible and competitive inhibition (Figure 10), which is consistent with other SGLT2 inhibitors [23,24]. However, the affinity to SGLT2 inhibition seemed to be different among the three SGLT2 inhibitors (Table 3). In addition, the recovery of SGLT2 transport activity following the pretreatment of DWP16001, dapagliflozin, and ipragliflozin for 24 h was retained at a higher concentration (200 nM) of DWP16001 compared with dapagliflozin and ipragliflozin. These results suggested that DWP16001 had the highest SGLT2 inhibition potential and that this inhibition potential retained for a longer time compared with dapagliflozin and ipragliflozin. Combined with the higher kidney distribution of DWP16001, retained SGLT2 inhibition with a high concentration of DWP16001 could also potentiate the efficacy of DWP16001 compared with dapagliflozin and ipragliflozin.

The comparative pharmacokinetics and in vitro SGLT2 inhibition findings suggest that DWP16001 might be a superior alternative to dapagliflozin and ipragliflozin; however, we should note that comparisons of the in vivo pharmacologic properties of these agents using therapeutic doses in animals and humans need to be further undertaken.

## Figures and Tables

**Figure 1 pharmaceutics-12-00268-f001:**
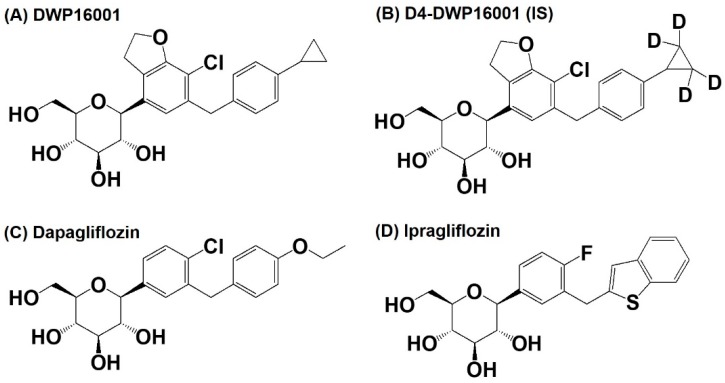
Structure of (**A**) DWP16001, (**B**) D4-DWP16001 (IS), (**C**) dapagliflozin, and (**D**) ipragliflozin.

**Figure 2 pharmaceutics-12-00268-f002:**
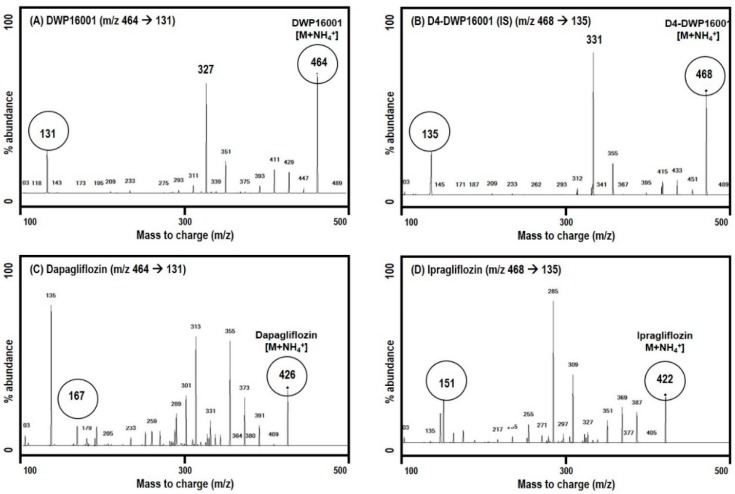
Product ion spectra of (**A**) DWP16001, (**B**) D4-DWP16001 (IS), (**C**) dapagliflozin, and (**D**) ipragliflozin.

**Figure 3 pharmaceutics-12-00268-f003:**
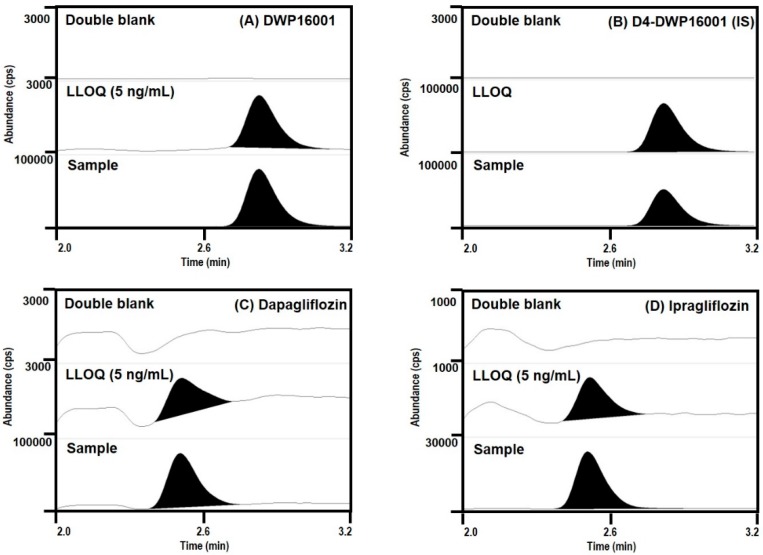
Representative multiple reaction-monitoring (MRM) chromatograms of (**A**) DWP16001, (**B**) D4-DWP16001 (IS), (**C**) dapagliflozin, and (**D**) ipragliflozin in mouse double-blank plasma, blank plasma spiked with DWP16001, dapagliflozin, ipragliflozin at the lower limit of quantification (LLOQ) (5 ng/mL), and plasma samples at 1 h following single oral administration of DWP16001, dapagliflozin, or ipragliflozin at a dose of 1 mg/kg.

**Figure 4 pharmaceutics-12-00268-f004:**
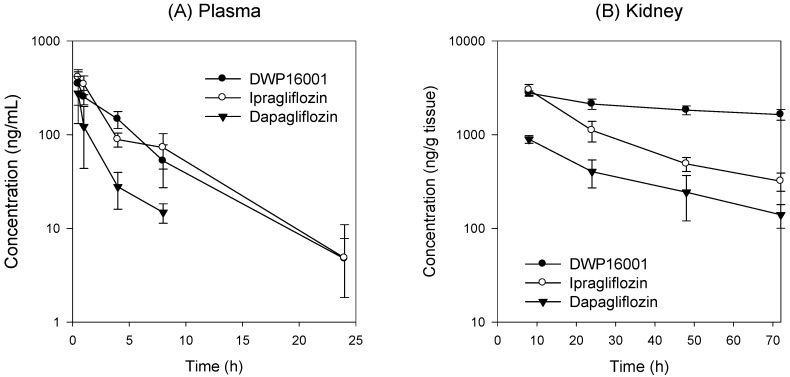
(**A**) Plasma and (**B**) kidney concentration vs. time profile of DWP16001 (●), dapagliflozin (▼), and ipragliflozin (○) after a single oral administrations of DWP16001, dapagliflozin, and ipragliflozin at a dose of 1 mg/kg in Institute of Cancer Research (ICR) mice, respectively. Data are expressed as mean±SD from five mice.

**Figure 5 pharmaceutics-12-00268-f005:**
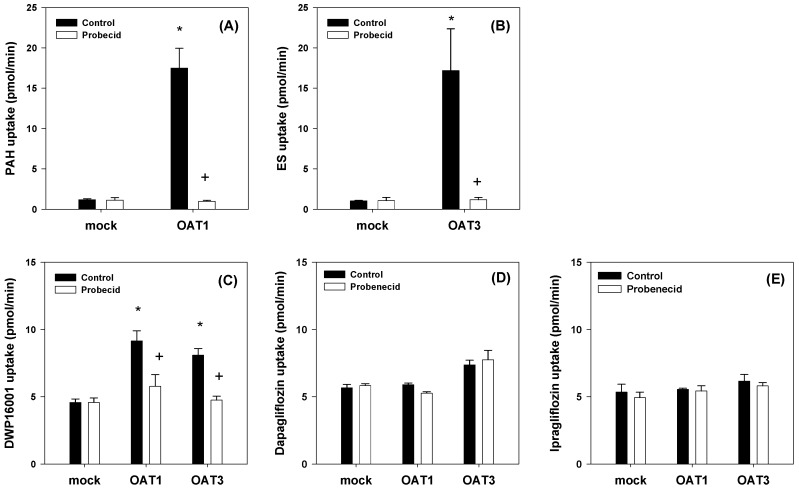
(**A**) Uptake of 0.1 μM [^3^H]para-aminohippuric acid (PAH) was measured in HEK293-mock and –OAT1 cells in the presence and absence of 20 μM probenecid for 5 min. (**B**) Uptake of 0.1 μM [^3^H]Estrone-3-sulfate (ES) was measured in HEK293-mock and –OAT3 cells in the presence and absence of 20 μM probenecid. Uptake of (**C**) DWP16001, (**D**) dapagliflozin, and (**E**) ipragliflozin (2 μM each) into HEK293-mock and HEK293 cells expressing OAT1 and OAT3 was measured for 5 min. Each data point represents the mean±standard deviation of triplicate experiments. *: *p* < 0.05, compared with mock cells; +: *p* < 0.05, compared with control group.

**Figure 6 pharmaceutics-12-00268-f006:**
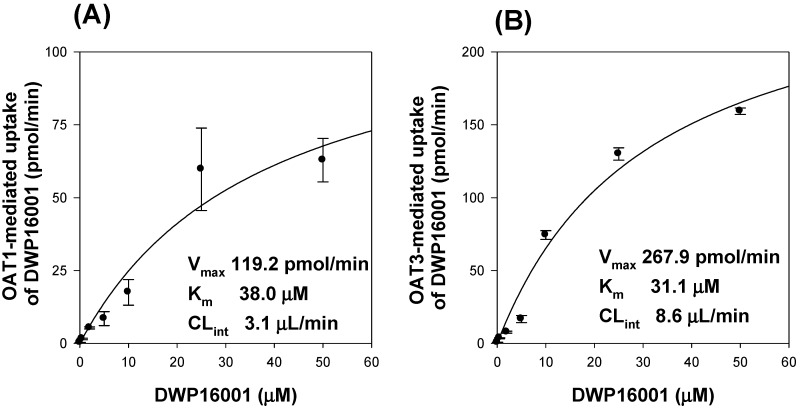
(**A**) OAT1- and (**B**) OAT3-mediated uptake of DWP16001. Concentration dependent uptake of DWP16001 was measured for 5 min into HEK293-mock cells and HEK293 cells expressing OAT1 and OAT3. The transporter-mediated uptake rate was obtained by subtracting the uptake in HEK293-mock cells (passive diffusion) from those in HEK293 cells expressing OAT1 and OAT3 (total uptake). Each data point represents the means ± standard deviation from triplicate experiments.

**Figure 7 pharmaceutics-12-00268-f007:**
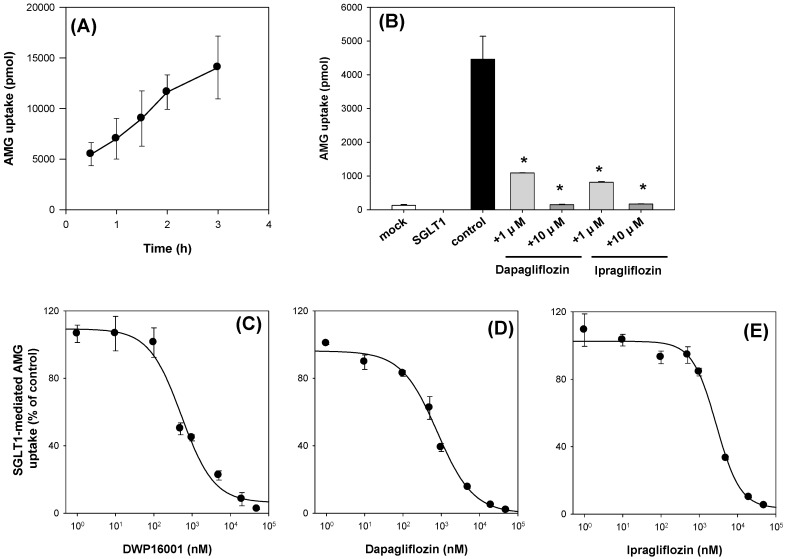
(**A**) Uptake of 10 μM [^14^C]methyl-a-D-glucopyranoside (AMG) was measured in CHO- sodium-glucose cotransporter 1 (SGLT1) cells with a various incubation time (0.5–3 h). (**B**) Inhibitory effect of dapagliflozin and ipragliflozin (1, 10 μM) on the uptake of 10 μM [^14^C]AMG in CHO-mock and -SGLT1 cells was measured for 2 h. Concentration dependent inhibition of DWP16001 (**C**), dapagliflozin (**D**), and ipragliflozin (**E**) on the SGLT1-mediated uptake of [^14^C]AMG. SGLT1-mediated uptake of [^14^C]AMG was calculated by subtracting the uptake of 10 μM [^14^C]AMG for 2 h in CHO-mock cells from that in CHO-SGLT1 cells in a concentration range of 1–50,000 nM of DWP16001, dapagliflozin, and ipragliflozin. Each data point represents the mean ± standard deviation of three independent experiments. *: *p* < 0.05, compared with control group.

**Figure 8 pharmaceutics-12-00268-f008:**
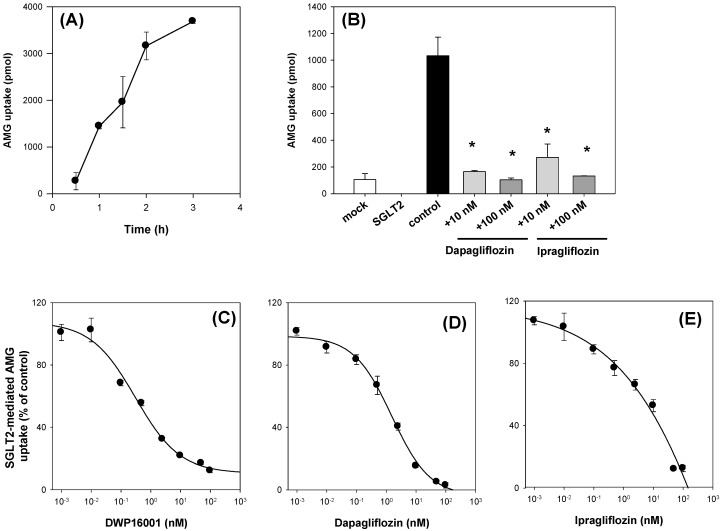
(**A**) The uptake of 10 μM [^14^C]methyl-a-D-glucopyranoside (AMG) was measured in CHO-SGLT2 cells with a various incubation time (0.5, 1, 1.5, 2, 3 h). (**B**) Inhibitory effect of dapagliflozin and ipragliflozin (10, 100 nM) on the uptake of 10 μM [^14^C]AMG in CHO-mock and –SGLT2 cells was measured for 2 h. Representative concentration dependent inhibition of DWP16001 (**C**), dapagliflozin (**D**), and ipragliflozin (**E**) on the SGLT2-mediated uptake of [^14^C]AMG. SGLT2-mediated uptake of [^14^C]AMG was calculated by subtracting the uptake of 10 μM [^14^C]AMG for 2 h in CHO-mock cells from that in CHO-SGLT2 cells in a concentration range of 1–100 nM of DWP16001, dapagliflozin, and ipragliflozin. Each data point represents the mean±standard deviation of three independent experiments. *: *p* < 0.05, compared with control group.

**Figure 9 pharmaceutics-12-00268-f009:**
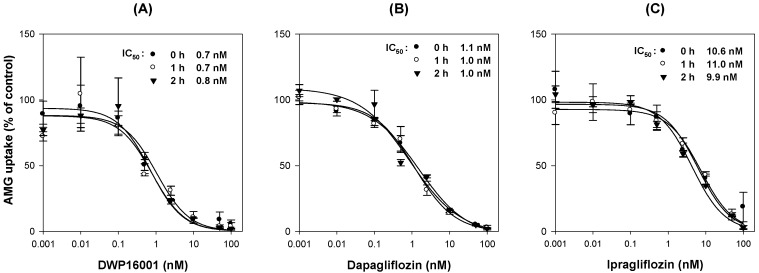
Inhibitory effect of (**A**) DWP16001, (**B**) dapagliflozin, and (**C**) ipragliflozin on the SGLT2-mediated uptake of 10 μM [^14^C]methyl-a-D-glucopyranoside (AMG) was measured with preincubation time for 1 and 2 h or without preincubation of DWP16001, dapagliflozin, and ipragliflozin, respectively, in a concentration range of 0.001–100 nM. Each data point represents the mean±standard deviation of three independent experiments.

**Figure 10 pharmaceutics-12-00268-f010:**
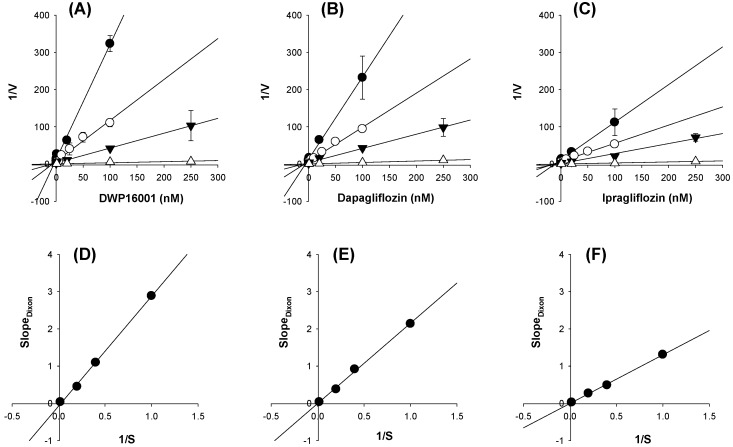
Dixon plot for the inhibitory effect of (**A**) DWP16001, (**B**) dapagliflozin, and (**C**) ipragliflozin on the uptake of [^14^C]methyl-a-D-glucopyranoside (AMG) in CHO-SGLT2 cells. Each symbol represents the concentration of [^14^C]AMG: ●, 1 μM; ○, 2.5 μM; ▼, 5 μM; △, 50 μM. Replot of the slopes of Dixon plot (slopes vs 1/[S]) was shown for (**D**) DWP16001, (**E**) dapagliflozin, and (**F**) ipragliflozin. V indicates the SGLT2-mediated uptake rate of [^14^C]AMG and S indicates the concentration of [^14^C]AMG. The data are the means±standard deviations of triplicate measurements.

**Figure 11 pharmaceutics-12-00268-f011:**
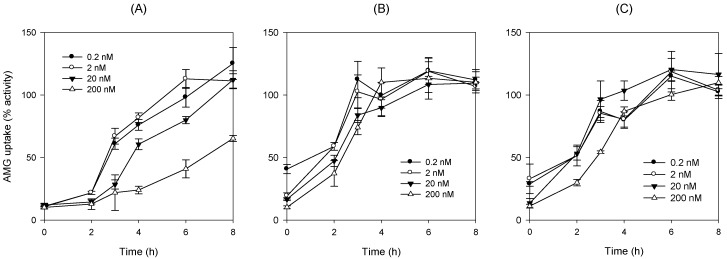
Recovery of SGLT2 transport function with different washout time following 24 h of incubation with (**A**) DWP16001, (**B**) dapagliflozin, and (**C**) ipragliflozin (0.2–200 nM) was measured through the uptake of [^14^C]methyl-a-D-glucopyranoside (AMG) in CHO-SGLT2 cells. Each data point represents the mean±standard deviation of three independent experiments.

**Table 1 pharmaceutics-12-00268-t001:** Pharmacokinetic parameters of DWP16001, dapagliflozin, and ipragliflozin after a single oral administrations of DWP16001, dapagliflozin, and ipragliflozin at a dose of 1 mg/kg in ICR mice, respectively.

	Parameters	DWP16001	Dapagliflozin	Ipragliflozin
Plasma	C_max_ (ng/mL)	371.4 ± 108.6	274.8 ± 143.6	409.6 ± 59.1
	T_max_ (h)	0.7 ± 0.3	0.5 ± 0.0	0.6 ± 0.2
	AUC_72h_ (μg∙h/mL)	1.69 ± 0.34	0.48 ± 0.18 *	1.96 ± 0.41
	AUC_∞_ (μg∙h/mL)	1.73 ± 0.34	0.55 ± 143.8 *	1.97 ± 0.41
	t_1/2_ (h)	3.8 ± 1.4	2.1 ± 0.2 *	3.1 ± 0.5
Kidney	AUC_72h_ (μg∙h/g tissue)	139.2 ± 5.19	26.30 ± 3.34 *	73.65 ± 7.05 *
	AUC ratio	85.0 ± 16.1	64.6 ± 31.8	38.4 ± 5.3 *
	t_1/2_ (h)	125.5 ± 80.4	24.9 ± 5.4 *	24.3 ± 5.7 *

Area under curve (AUC) ratio: Ratios of Kidney AUC to plasma AUC; Data expressed as mean ± SD from five mice; *: *p* < 0.05, compared with DWP16001 group.

**Table 2 pharmaceutics-12-00268-t002:** Protein binding of DWP16001, dapagliflozin, and ipragliflozin in mouse plasma and kidney homogenates.

Compounds	logP ^a^	Protein Binding (% Bound) ^b^	
		Plasma	Kidney
DWP16001	2.65	98.50 ± 0.20	97.24 ± 0.08
Dapagliflozin	2.27	94.19 ± 1.07	95.83 ± 0.68
Ipragliflozin	2.56	96.58 ± 0.75	96.50 ± 1.01

^a^ logP values were obtained from the partition coefficient of n-octanol/water; ^b^ Data expressed as mean ± SD of triplicate experiments.

**Table 3 pharmaceutics-12-00268-t003:** Inhibition potential of DWP16001, dapagliflozin, and ipragliflozin on SGLT1 and SGLT2 activities.

	IC_50_ (nM)		Selectivity
	SGLT1	SGLT2	(IC_50_ Ratio of SGLT1/SGLT2)
DWP16001	549.3 ± 139.6	0.8 ± 0.3	667
Ipragliflozin	2328.7 ± 377.6	8.9 ± 1.7	261
Dapagliflozin	803.0 ± 96.8	1.6 ± 0.3	493

Data were expressed as mean ± SD from triplicate measurement.

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
