# Peer review of "Comparative Pharmacokinetics and Pharmacodynamics of a Novel Sodium-Glucose Cotransporter 2 Inhibitor, DWP16001, with Dapagliflozin and Ipragliflozin"

_pharmaceutics, 2020, doi:10.3390/pharmaceutics12030268_

Round 1

Reviewer 1 Report

The present article is an elegant comparative study of a novel candidate for SGLT2-inhibitor. The rationale, methodology and discussion is  very clear and authors should be commended.

Author Response

Reviewer #1:

Comments: The present article is an elegant comparative study of a novel candidate for SGLT2-inhibitor. The rationale, methodology and discussion is very clear and authors should be commended.

Answer> We deeply appreciate the positive review on this manuscript.

Reviewer 2 Report

The authors of the manuscript entitled "Comparative pharmacokinetics and pharmacodynamics of a novel sodium-glucose cotransporter 2 inhibitor, DWP16001, with dapagliflozin and ipragliflozin" present decribe profound analysis of the properties of a novel drug candidate DWP16001. I generally find the manuscript very well written and intersting. The only minor corrections nedded are:

  • AUC is area under curve, not under concentration (line 20)
  • induced not examined (line 55)
  • changing the ions decriptions from [M-NH4+] to [M+NH4+], since the parent ions are adducts with ammionium cation, not the  products of the elimination of the ammonium cation (Fig. 2)

I recommend accepting the manuscript after minor corrections.

Author Response

Reviewer #2:

We deeply appreciate the positive review on this manuscript.

Q1. AUC is area under curve, not under concentration (line 20)

Answer> According to the reviewer’s comment, we changed “area under concentration” into “area under curve”

Q2. induced not examined (line 55)

Answer> According to the reviewer’s comment, we changed “examined” into “induced”

Q3. changing the ions decriptions from [M-NH4+] to [M+NH4+], since the parent ions are adducts with ammionium cation, not the  products of the elimination of the ammonium cation (Fig. 2)

Answer> According to the reviewer’s comment, we also changed [M-NH4+]” into “[M+NH4+]”

Reviewer 3 Report

The aim is to investigate the pharmacokinetics and kidney distribution of DWP16001, a novel SGLT2 inhibitor, and to compare these properties with those of dapagliflozin and ipragliflozin, representative SGLT2 inhibitors. DWP16001 showed the highest kidney distribution among three SGLT2 inhibitors when expressed as an area under concentration (AUC) ratio of kidney to plasma.

Comments

  1. Did you look at its lipophilicity or protein binding in each SGLT2 inhibitor? If possible, please show its volumes of distribution (Vd) or percentage of protein binding of each SGLT2 inhibitor.

Author Response

Q1. Did you look at its lipophilicity or protein binding in each SGLT2 inhibitor? If possible, please show its volumes of distribution (Vd) or percentage of protein binding of each SGLT2 inhibitor.

Answer> Thank you for your comments. As the reviewer suggested, we added logP values of three compounds in the Table 2, together with the protein binding results of three compounds in the plasma and kidney homogenates. And the results and interpretation was added in the result section as follows:

(page 9, line 296) The plasma and kidney tissue protein binding of DWP16001, dapagliflozin, and ipragliflozin were high and comparable and the logP values of three compound were also comparable (i.e. in the range of 2.27-2.65) (Table 2). It suggests that the large distribution of DWP16001, dapagliflozin, and ipragliflozin in the kidney was not necessarily reliant on plasma/tissue-specific binding ability or lipophilicity of these compounds.

However, we did not perform the pharmacokinetic studies of DWP16001, dapagliflozin, and ipragliflozin following intravenous injection. Therefore, we could not show volume of distribution of these three compounds here. Since we are still under investigation on the pharmacokinetics and metabolism of DWP16001, we are going to compare and discuss the relationship between the lipophilicity and the volume of distribution in the further studies.

We ask the reviewer's generous understanding on this issue.